# Holistic Modeling In Medical Image Segmentation Using Spatial Recurrence

**João B. S. Carvalho** [1]                    JOAO.CARVALHO@INF.ETHZ.CH
[1] *Department of Computer Science, ETH Zürich, Switzerland*
**João Santinha** [2,3]          JOAO.SANTINHA@RESEARCH.FCHAMPALIMAUD.ORG
[2] *IST, ULisboa, Portugal*
[3] *CCIG, Champalimaud Foundation, Portugal*
**Đorđe Miladinović** [1]                    DJORDJE.MILADINOVIC@INF.ETHZ.CH
**Carlos Cotrini** [1]                              CCARLOS@INF.ETHZ.CH
**Joachim M. Buhmann** [1]                          JBUHMANN@INF.ETHZ.CH

## Abstract

In clinical practice, regions of interest in medical imaging (MI) often need to be identified through a process of precise image segmentation. For MI segmentation to generalize, we need two components: to identify local descriptions, but at the same time to develop a holistic representation of the image that captures long-range spatial dependencies. Unfortunately, we demonstrate that the start of the art does not achieve the latter. In particular, it does not provide a modeling that yields a global, contextual model. To improve accuracy, and enable holistic modeling, we introduce a novel deep neural network architecture endowed with spatial recurrence. The implementation relies on gated recurrent units that directionally traverse the feature map, greatly increasing each layers receptive field and explicitly modeling non-adjacent relationships between pixels. Our method is evaluated in four different segmentation tasks: nuclei segmentation in microscopy images, colorectal polyp segmentation in colonoscopy videos, liver segmentation in abdominal CT scans, and aorta artery segmentation in thoracic CT scans. Our experiments demonstrate an average improvement in performance of 4.72 Dice points and 0.68 Hausdorff distance units comparing to U-Net and U-Net++, and a performance better or on par when compared to transformer-based architectures. Code available at https://github.com/JoaoCarv/holistic-seg.
**Keywords:** Medical Image Segmentation, U-Net, Spatially Recurrent Modeling

## 1. Introduction

Imaging the internal tissues of a patient can be crucial for the diagnosis, prognosis, and treatment planning (Son et al., 2021). Whether the medical imaging-driven patient assessment is done by medical professionals or automatic methods, a typical preceding step in the clinical pipeline is to perform image segmentation to reduce the dimensionality of the data and highlight regions of interest (Giger, 2018). Since manual segmentation is a time-consuming and tedious task for physicians, many algorithms have been developed to automate this process. The most successful recent approaches are based on encoder-decoder-based convolutional neural networks, stemming from the original U-Net architecture (Ronneberger et al., 2015).

For automatic medical image segmentation to achieve human-level performance it requires *holistic modeling*, critically relying on the capability of the machine learning model to both: *(i)* accurately identify *local* intensity discontinuities or edges as object boundaries

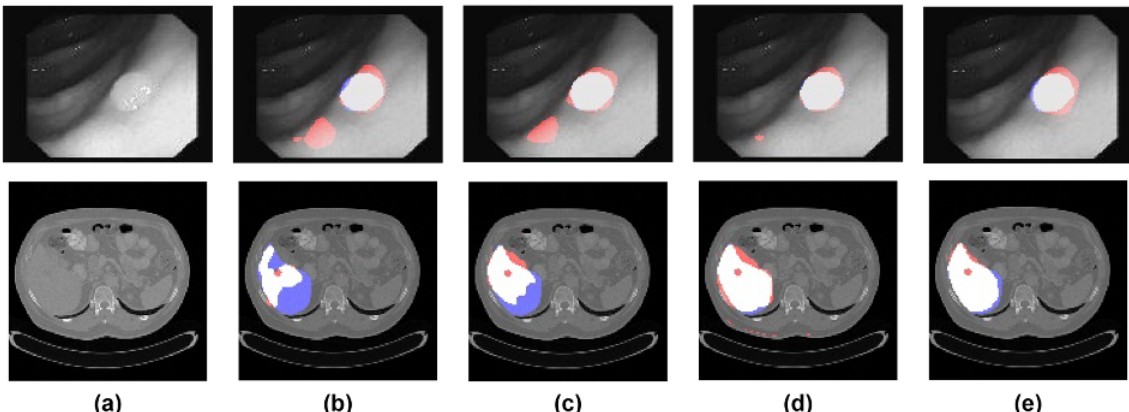

(a)    (b)    (c)    (d)    (e)

Figure 1: **Qualitative analysis of image segmentation for different competing methods (best viewed in color). (a)** original image; **(b)** U-Net (Ronneberger et al., 2015); **(c)** U-Net++ (Zhou et al., 2019); **(d)** SDU-Net (ours); **(e)** SDNU-Net (ours); **(top row)** *polyp segmentation*. The baseline U-Net architectures mistakenly identify a region in the bottom left corner as relevant due to the local features. However, when observing the image as a whole, it is clear that there is a single region of interest, as marked successfully by our networks. **(bottom row)** *liver segmentation*. In contrast to SDN-based networks, the baseline U-Net architectures are unable to coherently identify the well-shaped region of interest. **(red color)** false positives; **(blue color)** false negatives; **(white color)** correctly predicted pixels.

that characterize local textures; *(ii)* account for *global* contextual information when assessing the relevance of different image regions, e.g. understanding image-specific semantics and texture. For example, when segmenting a liver in a medical image, it is important to take into account both local texture patterns that describe the organ, as well as the contrasting characteristics of different regions, to model its consistent local description and global anatomical position. Convolutional neural networks excel at *(i)*, but often fail at *(ii)*, ultimately not achieving true holistic modeling.

We demonstrate this limitation in Figure 1 with two examples: colorectal polyp and liver segmentation, where commonly used architectures like U-Net and U-Net++ (Zhou et al., 2019) fail to correctly identify parts of the liver and also incorrectly believe benign tissue to be a polyp. Our experiments in Section 3.3 further confirm such limitations.

Thus, the question we address here is how to empower existing networks to perform holistic comparisons. To tackle this issue, we propose instead a novel approach by using *spatial recurrence* and implementing it in the form of gated recurrent units that traverse broader regions of the image in order to capture holistic features. Another advantage to this approach is the extensive work on recurrent units that can easily leveraged for sequence modeling. (Yu et al., 2019). In this context, we use the novel spatial dependency networks (SDNs) (Miladinovic et al., 2021), which have been recently used for generative image modeling, resulting in a new state-of-the-art variational autoencoder in several settings.

We experimentally demonstrate the benefit of bringing recurrence into the convolutional neural networks (CNNs) for medical image segmentation on four different tasks: *(i)* cell structure segmentation (Caicedo et al., 2019); *(ii and iii)* two anatomical structure segmen-

tation tasks (Bilic et al., 2019; Lambert et al., 2020); and *(iv)* abnormal tissue segmentation (Bernal et al., 2017), comprising three distinctive medical imaging methods (respectively, microscopic imaging, colonoscopy video, abdominal and thoracic CT-scan). In all of them, we observe an increase in performance when including spatial recurrence into the segmentation architectures, with an average increase of 4.72 Dice points and decrease of 0.68 Hausdorff distance units across all tasks when comparing to classical segmentation architectures. In particular, for the liver segmentation task, we go from $86.34\pm1.72$ Dice points to $94.72\pm1.71$, by solely enhancing U-Net with nested SDNs units. Additionally, our method also compares favourably on most settings with other recently proposed state-of-the-art segmentation architectures.

## Related Work

**Classical integration of multi-scales** – To achieve holistic segmentation, several CNN-based works have been proposed the integration of features at multi-scales (Kamnitsas et al., 2015), with other innovations following the use of atrous convolution layers (Chen et al., 2017), self-attention (Schlemper et al., 2019), and pyramid networks (Feng et al., 2020). Despite their success, learning global long-range spatial dependencies as still been observed as a persistent limitation Zhang et al. (2021).

**Transformer-based architectures** – More recently, motivated by their success in the computer vision field several transformer-based architectures (Dosovitskiy et al., 2020; Touvron et al., 2021) have been proposed to better integrate global context in medical image segmentation. These can be subdivided into architectures that combine transformer networks and CNNs (Chen et al., 2021; Zhang et al., 2021), and fully transformer base architectures(Cao et al., 2021; Lin et al., 2021). Unfortunately, transformer-based architectures are heavy GPU memory consumers and pre-training reliant on large non-medical datasets.

## 2. Bringing Spatial Recurrence Into Convolutional Neural Networks

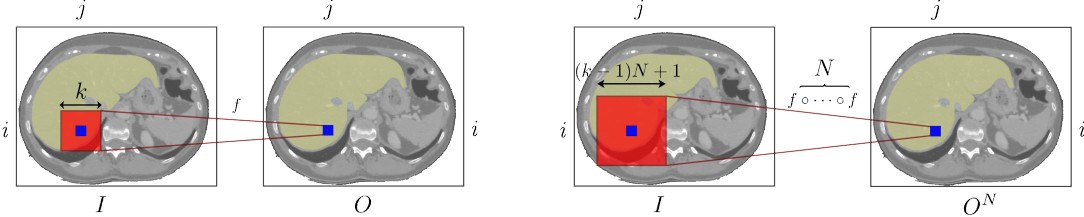

Figure 2: **Left:** Receptive field of a convolutional layer, with kernel size $k$. **Right:** Receptive field of $N$ convolutional layers with kernel size $k$ and stride 1.

We start with some auxiliary notation. Denote a 2D feature map as the tensor $X \in \mathbb{R}^{m \times n \times c}$, where $m$ is the feature map's width, $n$ is the feature map's height, and $c$ is the number of channels. For $i \leq m$ and $j \leq n$, we denote by $X_{i,j}$ the vector $(X_{i,j,1}, \ldots, X_{i,j,c})^{\top}$ that corresponds to the feature vector in position $(i,j)$. Observe that such description contemplates the 2D feature maps that are produced by convolutional layers and can be easily extended to the three-dimensional setting.

The receptive field of a convolutional layer often does not cover the entire image (Araujo et al., 2019). Although stacking such layers leads to a larger receptive field, as depicted Figure 2, this field may still be too small to cover the entire image (Araujo et al., 2019). For a more detailed presentation of the receptive fields, see Appendix A. In addition, such composition of layers only weakly models distant pixel relationships across different levels of the architecture. For example, consider a pixel inside the two red squares in Figure 2 and another one outside the smallest red square, but inside the largest one. The relationship between these two pixels is only perceived after composing at least two convolutional layers.

We argue that these limitations are unnecessarily restrictive for medical image segmentation. To address them, we propose to introduce recurrent sweeps across the feature map, thus involving more entries from $I$ through a spatially coherent modeling.

## 2.1. Recurrent Sweeps

Our proposal is to interleave *recurrent sweeps* with the convolutional layers. After computing the output $O \in \mathbb{R}^{m' \times n' \times c'}$ of a convolutional layer, we produce, using recurrent sweeps, another feature map $\hat{O}$ of same dimensions as $O$, which will be the input for the next layer.

We now show an example of a recurrent sweep that uses a recurrent unit $g_\downarrow$ to produce $\hat{O}$ from $O$ as follows. We assume a recurrent unit to be any function $g : \mathbb{R}^{c'} \times \mathbb{R}^{c'} \times \mathbb{R}^{c'} \times \mathbb{R}^{c'} \to \mathbb{R}^{c'}$. The recurrent sweep works as follows. For $i$ from 1 to $m$ and for $j$ from 1 to $n$, in that order, we compute

$$\hat{O}_{i,j} = g_\downarrow(O_{i,j}, \hat{O}_{i-1,j-1}, \hat{O}_{i-1,j}, \hat{O}_{i-1,j+1}) \qquad \text{(Figure 3).} \tag{1}$$

We call such a recurrent sweep a *downward sweep*. Figure 3 illustrates the features that influence the outcome of a downward sweep for one pixel in $\hat{O}_{i,j}$. In a similar way, we define sweeps in the other three directions: up, down, and left. A sweep can also use different types of recurrent units. For example, one that instead of $\{O_{i,j}, \hat{O}_{i-1,j-1}, \hat{O}_{i-1,j}, \hat{O}_{i-1,j+1}\}$ uses $\{O_{i,j}\} \cup \{\hat{O}_{r,s} : i - 2 \leq r \leq i, j - 1 \leq s \leq j + 1\}$.

A recurrent sweep can also be a composition of other sweeps. For example, we can first apply a downward sweep to $O$. Then successively apply a leftward sweep, an upward sweep, and a rightward to the outcome of each previous sweep. The output of this composite sweep for a particular entry of $\hat{O}$ involves then all entries in $O$ (Figure 3). We formally demonstrate this in Appendix A. The receptive field in this case is then the entire image. This is an advantage over convolutional layers, where the receptive field may not cover the entire image, even after stacking several convolutional layers.

## 2.2. Spatial Recurrence Through Spatially Dependent Networks

In this work we make use of the recently proposed spatial dependency layers (Miladinovic et al., 2021) to introduce spatial recurrence. They take as input a feature map $O$ and produce an output feature map $\hat{O}$ in three steps, as follows.
*Project-in stage:* This stage applies an affine transformation to $O$, yielding $\bar{O}$ as follows:

$$\bar{O}_{i,j} = O_{i,j}\mathbf{W} + \mathbf{b}, \tag{2}$$

where $\mathbf{W}$ and $\mathbf{b}$ are a learnable weight matrix and a learnable bias vector, respectively. $\bar{O}$ usually contains a larger and tunable number of channels than $O$.

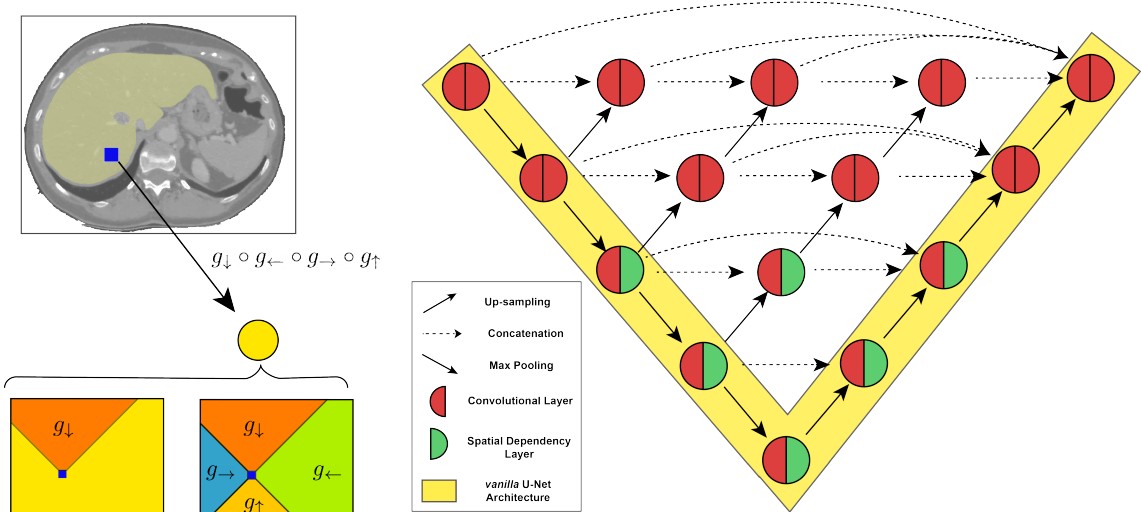

Figure 3: **Left: Receptive field.** Using recurrent sweeps it is possible to greatly extend the receptive field of a layer, as shown with the downward sweep, $g_\downarrow$. In particular, the composition of sweeps, $g$, covers the full input $O$, and, contrarily to convolutional layers, it is able to explicitly model long-range dependencies within the same feature map. **Right: Diagram of architectures.** SDNU-Net integrates a mix of convolutional and spatial dependency layers into the deeper scales of U-Net++. In yellow we depict SDU-Net.

*Correction stage:* This stage performs four recurrent sweeps over $\bar{O}$ in different directions, one after the other as explained in Section 2.2. They use a gating mechanism (Cho et al., 2014), adapted to the image setting, which moderates the contributions of the updated *(proposed)* value and the intermediate *(prior)* feature value.

*Project-out stage:* This stage performs an affine transformation on the output from the correction stage, yielding $\hat{O}$. The number of channels in $\hat{O}$ equals those in $O$.

Note that a more detailed description of each step of a SDN layer and the algorithmic formalization of the correction stage can be found in Appendix B.

### 2.3. Introducing Spatial Dependency Layers to U-Net and U-Net++

Our main contribution is the integration of recurrence into convolutional neural networks. We illustrate this by introducing spatial dependency layers into U-Net and U-Net++, which are the backbones for many of the best performant segmentation networks (Jha et al., 2021; Srivastava et al., 2021) (see Figure 3). U-Net++, by incorporating nested dense connections to the U-Net architecture, mainly attempts to alleviate the drawbacks of features from different semantic levels being combined at the decoding path. Following our ablation studies (Figure 4), we found it sufficient to include spatial dependency layers at lower scales of the U-Net++, retaining most of the original nested architecture, and mitigating the computational complexity inherent to SDNs. This was also verified for the vanilla U-Net. Our implementation consecutively applies convolutional and spatial dependency layers within the same depth level.

## 3. Experiments

### 3.1. Datasets

As described in Table 1, four public medical image segmentation datasets were included in our study, covering three different medical imaging modalities and four segmentation tasks. Partitioning between train, validation, and test sets was performed at the volume/patient level to avoid intra-sample bias and over-optimistic results. For further details on the datasets and corresponding data pre-processing pipelines, please refer to Appendix C.

Table 1: Summary information of datasets used.

|  | Number of Images | Image Size (resampled size) | Modality | Challenge |
|---|---|---|---|---|
| **Nuclei** | 670 (2D images) | 96×96 (no resampling) | Microscopy | 2018 Data Science Bowl (Caicedo et al., 2019) |
| **Polyps** | 612 (29 sequences) | $384 \times 288$ ($192 \times 144$) | Colonoscopy | Endoscopic Vision MICCAI 2015 (Bernal et al., 2017) |
| **Liver** | 131 (3D volumes) | $512 \times 512$ ($128 \times 128$) | CT | LiTS ISBI 2016/MICCAI 2017 (Bilic et al., 2019) |
| **Aorta** | 40 (3D volumes) | $256 \times 128$ (no resampling) | CT | Segthor ISBI 2019 (Lambert et al., 2020) |

### 3.2. Implementation Details

Following (Zhou et al., 2019; Isensee et al., 2021), the number of layers of the baseline network architecture was tuned to each segmentation task. Similarly, SDN specific parameters - state size, i.e. the number of channels in each *project-in stage* of spatial dependency layer, then number of directions, and the number of layers equipped with spatial dependency were separately optimized. Architecture details of all models, including activation functions and kernel sizes, followed original descriptions. Final configurations and implementation details are disclosed in Appendix D.

### 3.3. Results and Discussion

Table 2 benchmarks SDU-Net and SDNU-Net, as well as baselines, U-Net and U-Net++, in terms of segmentation performance measured in Dice index and Hausdorff distance (mean±s.d. across 5 folds). We also evaluated the performance of three other state-of-the-art methods: two transformer-based methods, (1) the CNN-transformer hybrid method, Trans-Unet, and (2) the fully transformer-based architecture, Swin-Unet, and (3) the nnU-Net, a widely used framework for automating U-Net pipeline's decisions, from which we derived a 3D model when the task allowed. For the sake of completeness, the Jaccard index is also included.

**Comparison with baselines** – Globally, the inclusion of spatial dependency layers improves model performance in all evaluated segmentation tasks, with both SDU-Net and SDNU-Net obtaining enhanced performance compared to its baselines. This coincides with an average increase of 4.72 Dice points and a decrease of the Hausdorff distance in 0.69 across all tasks and models. A more detailed summary and analysis of the performance improvements can be found in Appendix E.

Table 2: 5-fold cross-validation evaluation of baseline (U-Net and U-Net++), state-of-the-art (Trans-Unet, Swin-Unet and nnU-Net), and proposed (SDU-Net and SDNU-Net) models.

| | Dice index (↑) | Jaccard index (↑) | Hausdorff dist. (↓) |
|---|---|---|---|
| *Nuclei (Caicedo et al., 2019)* | | | |
| U-Net (baseline) | 87.20±0.89 | 77.30±0.79 | 1.89±0.43 |
| U-Net++ (baseline) | 89.67±0.62 | 81.27±0.58 | 1.39±0.34 |
| Trans-Unet (Chen et al., 2021) | 92.15±0.51 | 85.44±0.46 | 0.67±0.38 |
| Swin-Unet (Cao et al., 2021) | 92.21±0.51 | 85.55±0.43 | 0.72±0.34 |
| nnU-Net 2D (Isensee et al., 2021) | 94.20±0.40 | 89.04±0.30 | 0.50±0.31 |
| SDU-Net (ours) | 91.82±0.60 | 88.36±0.65 | 0.97±0.37 |
| SDNU-Net (ours) | 93.70±0.47 | 91.75±0.41 | 0.52±0.33 |
| *Polyps (Bernal et al., 2017)* | | | |
| U-Net (baseline) | 76.23±1.24 | 61.56±1.24 | 3.81±0.61 |
| U-Net++ (baseline) | 78.43±1.75 | 64.38±1.57 | 2.94±0.72 |
| Trans-Unet (Chen et al., 2021) | 83.85±1.30 | 72.19±1.25 | 2.79±0.61 |
| Swin-Unet (Cao et al., 2021) | 84.21±1.71 | 72.73±1.56 | 2.65±0.53 |
| nnU-Net 2D (Isensee et al., 2021) | 81.77±1.58 | 69.16±1.40 | 2.91±0.71 |
| SDU-Net (ours) | 82.30±1.57 | 69.92±1.36 | 2.72±0.54 |
| SDNU-Net (ours) | 85.14±1.80 | 74.13±1.61 | 2.03±0.63 |
| *Liver (Bilic et al., 2019)* | | | |
| U-Net (baseline) | 86.34±1.72 | 75.96±1.67 | 1.32±0.59 |
| U-Net++ (baseline) | 88.78±1.48 | 79.82±1.47 | 1.02±0.65 |
| Trans-Unet (Chen et al., 2021) | 94.32±1.51 | 89.25±1.36 | 0.71±0.41 |
| Swin-Unet (Cao et al., 2021) | 94.57±1.73 | 89.70±1.47 | 0.61±0.43 |
| nnU-Net 2D (Isensee et al., 2021) | 92.56±1.55 | 86.15±1.28 | 0.91±0.61 |
| nnU-Net 3D (Isensee et al., 2021) | 95.43±1.38 | 91.26±1.37 | 0.29±0.53 |
| SDU-Net (ours) | 93.21±1.44 | 87.28±1.37 | 0.83±0.51 |
| SDNU-Net (ours) | 94.72±1.71 | 89.97±1.43 | 0.67±0.42 |
| *Aorta (Lambert et al., 2020)* | | | |
| U-Net (baseline) | 90.76±0.93 | 83.08±2.78 | 0.68±0.32 |
| U-Net++ (baseline) | 92.96±0.78 | 86.85±1.12 | 0.61±0.27 |
| Trans-Unet (Chen et al., 2021) | 93.90±0.73 | 88.50±0.67 | 0.30±0.27 |
| Swin-Unet (Cao et al., 2021) | 94.20±0.81 | 89.04±0.71 | 0.21±0.31 |
| nnU-Net 2D (Isensee et al., 2021) | 92.20±0.79 | 85.53±0.60 | 0.35±0.41 |
| nnU-Net 3D (Isensee et al., 2021) | 92.60±0.87 | 86.22±0.74 | 0.38±0.32 |
| SDU-Net (ours) | 93.13±0.89 | 87.14±1.31 | 0.31±0.29 |
| SDNU-Net (ours) | 94.13±0.84 | 88.91±1.26 | 0.23±0.19 |

**Comparison with state-of-the-art methods** – In all segmentation tasks assessed the SDNU-Net architecture either outperforms or performs on par with transformer-based architectures. The nnU-Net automating pipeline still outperforms all models in the nuclei segmentation task, yet this is arguably a task where long-range dependencies play a diminished role. In all other segmentation tasks, the inclusion of long-range dependencies leads to improved performance when compared to nnU-Net 2D.

**Model stability** – To evaluate the stability of our architectures, each model's performance was also assessed across different initialization seeds. This study can be found in Appendix F.

**Scalability to higher dimensions** – To address scalability to larger and higher dimensional images the number of parameters and computational complexity was evaluated for SDN and corresponding baseline models. We verify that SDN models are within the same order of magnitude and scale similarly as the input size increases (Appendix G). The correction stage itself is $\mathcal{O}(N)$, with N being the output feature map scale, leading to higher GPU memory requirements when training. Despite the aforementioned computational consideration, when scaling to higher dimensions, and contrary to transformer-based architectures, our method offers the flexibility of adapting the number of sweeps and directions used, lowering its overall memory and computational overhead.

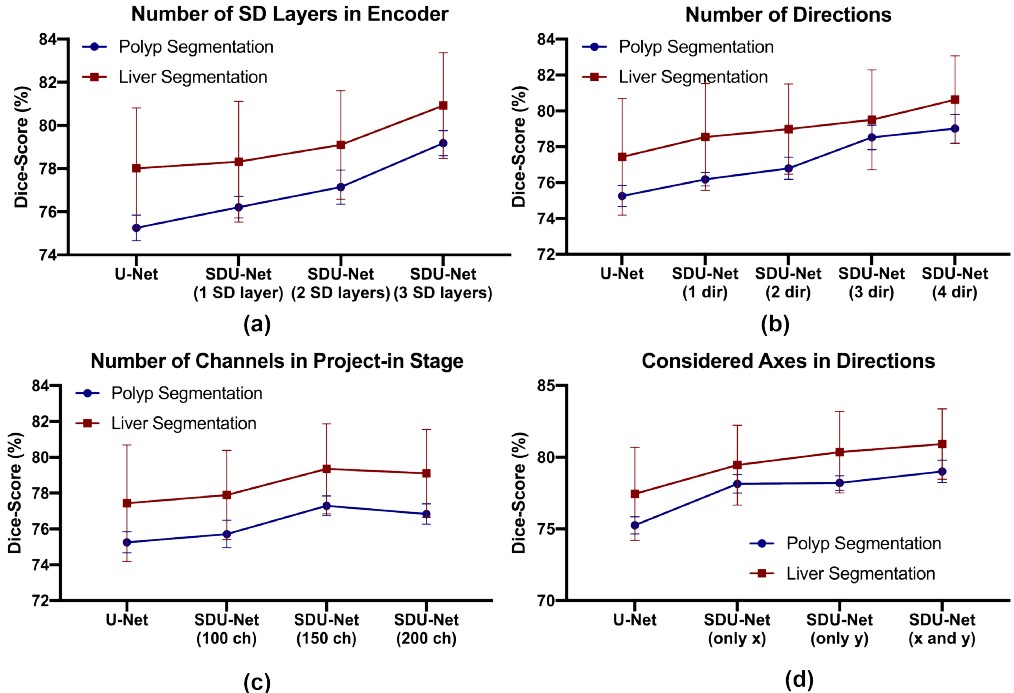

Figure 4: **Ablation studies.** Fixing SDU-Net with 1 spatial dependency layer in the encoder, direction left-to-right, and 100 output channels in the *project-in stage*, all hyper-parameters were scanned, and models compared to a 4 layer U-Net.

**Ablation studies** – Fixing the general design of the architectures, and evaluating in the colon polyp segmentation and liver segmentation tasks, contributions from SD specific parameters were estimated. We verify that increased number of scales equipped with spatial dependency layers (Figure 4 a), number of directions (Figure 4 b), and number of output channels in the project-in stage (Figure 4 c) lead to an overall improvement in performance. The experiments additionally suggest that the choice of sweeping directions (Figure 4 d) also impacts performance. For both segmentation tasks, performing two sweeps across two different axes is preferential to sweeping bidirectionally across the same axis.

## 4. Conclusion

In this work we motivated holistic modeling for automatic segmentation of medical images and proposed ways to bridge the gap by introducing spatial recurrence into convolutional neural networks. In order to achieve this goal, we designed two novel SDN based architectures (SDNU-Net and SDU-Net) that greatly increase the receptive field of CNNs, while explicitly modeling long-range dependencies in the feature map. Through experiments in four segmentation tasks (nuclei segmentation in microscopy images, colorectal polyp segmentation in colonoscopy videos, liver segmentation in abdominal CT scans, and aorta segmentation in thoracic CT scans) we have demonstrated the superior performance of both models. Ultimately, these are broadly generalizable architectures due the their inherent simple integration into U-Nets, and we believe can be widely adopted in this domain.

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

## Appendix A. Receptive field analysis

We present here how input features affect output features in convolutional layers and SDN layers. In particular, we compute *receptive fields* (Araujo et al., 2019) for these layers. A receptive field maps each feature output by a layer to a region in the input that defines that feature. Our goal is to demonstrate that the receptive field of one SDN layer covers the entire image (see Figure 3), whereas the receptive field of one convolutional layer may cover only a fraction of the image (see Figure 2). As a result, SDN layers capture long-range dependencies in the image, in comparison, to convolutional layers. We demonstrate how enhancing convolutional layers with SDN layers leads to higher performance in our experimental comparison in Section 3.

### A.1. Setup

The notation, presentation, and results given in this appendix resemble very much those from (Araujo et al., 2019). Assume given a convolutional network with $L$ layers, $l = 1, 2, \ldots, L$. Define a feature map $x_l \in \mathbb{R}^{h_l \times w_l \times d_l}$ to denote the output of the $l$-th layer, where $h_l, w_l$, and $d_l$ denote the output's height, width, and depth. We denote the input image by $x_0$ and the feature map output by the last layer as $x_l$. We consider layers whose output depends locally on input features, like convolution and pooling.

We restrict the presentation, without loss of generality, to one dimension. Each layer $l \leq L$ is parameterized by 4 variables: the kernel size $k_l > 0$, the stride $s_l > 0$, the left padding $p_l \geq 0$, and the right padding $q_l \geq 0$.

### A.2. Receptive field size for convolutional and pooling layers

We follow closely the presentation from Araujo et al. (2019) for this part. For $l \leq L$, we now compute the *receptive field size $r_l$ of $x_L$ with respect to $x_l$*. This is the number of features in $x_l$ which contribute to generating one feature in $x_L$. We do this inductively, from $r_L$ to $r_1$. For convenience, we define $r_L$ as 1.

Assume then that $r_l$ has been computed. We now compute $r_{l-1}$ for the case where $p_l = q_l = 0$ and $k_l = 1$. Here, the $r_l$ features in $f_l$ will cover $r_{l-1} = s_l r_l - (s_l - 1)$ features in $f_{l-1}$. More generally, when $k_l > 1$,

$$r_{l-1} = s_l r_l + (k_l - s_l), \text{ for } l \leq L. \tag{3}$$

Solving this recurrence equation for $r_0$ yields (Araujo et al., 2019)

$$r_0 = 1 + \sum_{l=1}^{L} \left( (k_l - 1) \prod_{i=1}^{l-1} s_i \right). \tag{4}$$

As Araujo et al. (2019) remark, this equation makes intuitive sense for some special cases. If $k_l = 1$, for all $l$, then $r_0 = 1$. If $s_l = 1$, for all $l$, then $r_0 = 1 + \sum_{l=1}^{L} (k_l - 1)$.

We define the *receptive field of a feature of $x_L$, with respect to $x_l$* as the region in $x_l$ that generated that feature (Araujo et al., 2019). This field can be defined as all features in $x_l$ between a particular left-most feature $u_l$ and a particular right-most feature $v_l$. Araujo

et al. (2019) demonstrate that

$$u_{l-1} = -p_l + u_l\, s_l \qquad\qquad \text{for } l \leq L \text{ and} \qquad (5)$$

$$v_{l-1} = -p_l + v_l\, s_l + k_l - 1 \qquad\qquad \text{for } l \leq L. \qquad (6)$$

Solving these recurrence equations for $u_0$ and $v_0$ yields (Araujo et al., 2019)

$$u_0 = u_L \prod_{i=1}^{L} s_i - \sum_{l=1}^{L} p_l \prod_{i=1}^{l-1} s_i \qquad (7)$$

$$v_0 = v_L \prod_{i=1}^{L} s_i - \sum_{l=1}^{L} (1 + p_l - k_l) \prod_{i=1}^{l-1} s_i. \qquad (8)$$

The set of features between $u_0$ and $v_0$ is exactly all the features that influence the outcome of one feature in $f_L$. Observe that by increasing $L$, $u_0$ moves to the left whereas $v_0$ moves to the right.

### A.3. Receptive field for SDN layers

For this part, we use the notation from Section 2.2. Recall that $O$ and $\hat{O}$, both in $\mathbb{R}^{m' \times n' \times c'}$, are the input and output of an SDN layer.

We argue here that the receptive field of $\hat{O}_{h,k}$ with respect to $O$ is the entire $O$, for $h \leq m'$ and $k \leq n'$. We formalize this in the theorem below.

**Theorem 1** *For any $i \leq m'$ and $j \leq n'$, $O_{i,j}$ influences the outcome of $\hat{O}_{h,k}$.*

For the proof, we need the following three lemmas.

**Lemma 2** *For the project-in stage, the receptive field of $\bar{O}_{h,k}$ with respect to $O$ is $O_{h,k}$, for $h \leq m'$ and $k \leq n'$.*

**Lemma 3** *For the correction stage, the receptive field of $\tilde{O}_{h,k}$ with respect to $\bar{O}$ is the entire feature map $\bar{O}$, for $h \leq m'$ and $k \leq n'$.*

**Lemma 4** *For the project-out stage, the receptive field of $\hat{O}_{h,k}$ with respect to $\tilde{O}$ is $\tilde{O}_{h,k}$, for $h \leq m'$ and $k \leq n'$.*

The proofs of Lemmas 2 and 4 follows from the fact that each stages applies an affine transformation to each entry of their corresponding input.

The proof of Lemma 3 consists of showing that for any $i, h \leq m'$ and $j, k \leq n'$, $\bar{O}_{i,j}$ must be in one of the four triangles depicted in Figure 3 and that, for each of these four triangles, any entry in that triangle influences $\tilde{O}_{h,k}$.

**Lemma 5** *For $i, h \leq m'$ and $j, k \leq n'$. if $i \leq h$ and $|j - k| \leq h - i$, then $\bar{O}_{i,j}$ and $\tilde{O}_{i,j}^{\downarrow}$ influence the outcome of $\tilde{O}_{h,k}$.*

**Proof** The proof is by induction on $h - i$. For the base case when it is zero, observe that $\bar{O}_{i,j}$ influences $\tilde{O}_{i,j}^{\downarrow}$ in the first loop of Algorithm 1. $\tilde{O}_{i,j}^{\downarrow}$ influences $\tilde{O}_{i,j}^{\leftarrow}$ in the second loop of Algorithm 1. $\tilde{O}_{i,j}^{\leftarrow}$ influences $\tilde{O}_{i,j}^{\uparrow}$ in the third loop of Algorithm 1. $\tilde{O}_{i,j}^{\uparrow}$ influences $\tilde{O}_{i,j}^{\rightarrow}$ in the last loop. But $\tilde{O}_{i,j}^{\rightarrow} = \tilde{O}_{i,j} = \tilde{O}_{h,k}$, by the last line of Algorithm 1.

For the inductive case, suppose that $h - i > 0$. Suppose that $j > k$; the proof when $j < k$ or $j = k$ is analogous. We can instantiate the induction hypothesis with $i+1, h, j-1$, and $k$. Indeed, $h \geq i+1$ and $h-(i+1) = h-i-1 \geq |j-k|-1 = j-k-1 = (j-1)-k = |(j-1)-k|$. So, by the induction hypothesis, $\bar{O}_{i+1,j-1}$ and $\tilde{O}_{i+1,j-1}^{\downarrow}$ influence the outcome of $\tilde{O}_{h,k}$. In addition, $\bar{O}_{i,j}$ influences the outcome of $\tilde{O}_{i,j}$ and $\tilde{O}_{i,j}$ influences the outcome of $\tilde{O}_{i+1,j-1}$, by the first loop of Algorithm 1. We conclude then that $\bar{O}_{i,j}$ and $\tilde{O}_{i,j}$ influences the outcome of $\tilde{O}_{h,k}$, which is what we wanted to prove. ∎

The remaining lemmas have similar proofs.

**Lemma 6** *For $i, h \leq m'$ and $j, k \leq n'$, if $k \leq j$ and $|h - i| \leq k - j$, then $\bar{O}_{i,j}$ and $\tilde{O}_{i,j}^{\leftarrow}$ influence the outcome of $\tilde{O}_{h,k}$.*

**Lemma 7** *For $i, h \leq m'$ and $j, k \leq n'$, if $h \leq i$ and $|j - k| \leq i - h$, then $\bar{O}_{i,j}$ and $\tilde{O}_{i,j}^{\uparrow}$ influence the outcome of $\tilde{O}_{h,k}$.*

**Lemma 8** *For $i, h \leq m'$ and $j, k \leq n'$, if $j \leq k$ and $|h - i| \leq k - j$, then $\bar{O}_{i,j}$ and $\tilde{O}_{i,j}^{\rightarrow}$ influence the outcome of $\tilde{O}_{h,k}$.*

We now prove Lemma 3. That is, for $i, h \leq m'$ and $j, k \leq n'$, $\bar{O}_{i,j}$ influences the outcome of $\tilde{O}_{h,k}$. Observe that at least one of the following four conditions holds:

- $i \leq h$ and $|j - k| \leq h - i$.

- $k \leq j$ and $|h - i| \leq k - j$.

- $h \leq i$ and $|j - k| \leq i - h$.

- $j \leq k$ and $|h - i| \leq k - j$.

Each of these conditions implies, by the four lemmas above, that $\bar{O}_{i,j}$ influences the outcome of $\tilde{O}_{h,k}$.

## Appendix B. Formalization of the SDN layer

In this section we formalize how a spatial dependency layer works.

Let $O \in \mathbb{R}^{m' \times n' \times c'}$ be a feature map, where $m'$, $n'$, and $c'$ represent the width, height, and number of channels. An SDN produces a new output feature map $\hat{O} \in \mathbb{R}^{m' \times n' \times c'}$ in three stages.

*Project-in stage:* This stage applies an affine transformation to $O$, yielding $\bar{O}$ as follows:

$$\bar{O}_{i,j} = O_{i,j}\mathbf{W} + \mathbf{b}, \tag{9}$$

where $\mathbf{W} \in \mathbb{R}^{c' \times \bar{c}}$ and $\mathbf{b} \in \mathbb{R}^{\bar{c}}$ are a learnable weight matrix and a learnable bias vector, respectively. Usually, $\bar{c} > c'$. That is, $\bar{O}$ usually contains a larger and tunable number of channels than $O$, usually larger than the number of channels in $O$.

*Correction stage:* This stage performs four recurrent sweeps over $\bar{O}$ in different directions, as described in Section 2.1. Each sweep uses a gating mechanism (Cho et al., 2014), adapted to the image setting, which moderates the contributions of the updated *(proposed)* value and the intermediate *(prior)* feature value. A gating mechanism can be modeled as a function $g : \mathbb{R}^{\bar{c}} \times \mathbb{R}^{\bar{c}} \times \mathbb{R}^{\bar{c}} \times \mathbb{R}^{\bar{c}} \to \mathbb{R}^{\bar{c}}$. The four sweeps use four gating mechanisms $g_{\leftarrow}, g_{\uparrow}, g_{\rightarrow}$, and $g_{\downarrow}$. Algorithm 1 explains in detail how these four sweeps work and how they are combined to produce a feature map $\tilde{O}$.

*Project-out stage:* This stage performs an affine transformation on $\tilde{O}$, yielding $\hat{O} \in \mathbb{R}^{m' \times n' \times c'}$. The number of channels in $\hat{O}$ equals those in $O$.

---

**Algorithm 1:** Correction stage

---

Initialize a feature map $\tilde{O}^{\downarrow} \in \mathbb{R}^{m' \times n' \times \bar{c}}$.
**for** $i = 1, \ldots, m'$ **do**
 **for** $j = 1, \ldots, n'$ **do**
  $\tilde{O}^{\downarrow}_{i,j} = g_{\downarrow}(\bar{O}_{i,j}, \tilde{O}^{\downarrow}_{i-1,j-1}, \tilde{O}^{\downarrow}_{i-1,j}, \tilde{O}^{\downarrow}_{i-1,j+1})$
 **end**
**end**
Initialize a feature map $\tilde{O}^{\leftarrow} \in \mathbb{R}^{m' \times n' \times \bar{c}}$.
**for** $j = 1, \ldots, n'$ **do**
 **for** $i = m', \ldots, 1$ **do**
  $\tilde{O}^{\leftarrow}_{i,j} = g_{\leftarrow}(\tilde{O}^{\downarrow}_{i,j}, \tilde{O}^{\leftarrow}_{i-1,j+1}, \tilde{O}^{\leftarrow}_{i,j+1}, \tilde{O}^{\leftarrow}_{i+1,j+1})$
 **end**
**end**
Initialize a feature map $\tilde{O}^{\uparrow} \in \mathbb{R}^{m' \times n' \times \bar{c}}$.
**for** $i = m', \ldots, 1$ **do**
 **for** $j = n', \ldots, 1$ **do**
  $\tilde{O}^{\uparrow}_{i,j} = g_{\uparrow}(\tilde{O}^{\leftarrow}_{i,j}, \tilde{O}^{\uparrow}_{i+1,j+1}, \tilde{O}^{\uparrow}_{i+1,j}, \tilde{O}^{\uparrow}_{i+1,j-1})$
 **end**
**end**
Initialize a feature map $\tilde{O}^{\rightarrow} \in \mathbb{R}^{m' \times n' \times \bar{c}}$.
**for** $j = 1, \ldots, m'$ **do**
 **for** $i = 1, \ldots, n'$ **do**
  $\tilde{O}^{\rightarrow}_{i,j} = g_{\rightarrow}(\tilde{O}^{\uparrow}_{i,j}, \tilde{O}^{\rightarrow}_{i-1,j-1}, \tilde{O}^{\rightarrow}_{i,j-1}, \tilde{O}^{\rightarrow}_{i+1,j-1})$
 **end**
**end**
Define $\tilde{O}$ as $\tilde{O}^{\rightarrow}$.

---

## Appendix C. Dataset details

Table 3: Relevant dataset statistics and main pre-processing steps.

| Segmentation Task | *Nuclei* | *Polyps* | *Liver* | *Aorta* |
|---|---|---|---|---|
| Dataset name | DSB2018 | CVC-ClinicDB | LiTS | SegTHOR |
| Image type | brightfield and fluorescent microscopy | colonoscopy video | abdominal CT scan | chest CT scan |
| Dataset size | 670 images | 29 sequences (612 frames) | 131 volumes | 40 volumes |
| Original size | 96 x 96 | 304 x 288 | 512 x 512 | 256 x 128 |
| Resampled size | - | 192 x 144 | 128 x 128 | not resampled |
| Data split for stability analysis[2] | 80/10/10 (%) | 23/3/3 (sequences) | 82/21/28 [3] (volumes) | 32/4/4 (volumes) |
| Intensity clipping | - | - | [-1000; 1000] HU | [-1000; 1000] HU |
| Rescale to [0,1] | Yes | Yes | Yes | Yes |

[1] Partition at sequence level for CVC-ClinicDB and at the volume/patient level otherwise

[2] From the two original available training batches, one (103 volumes) was split 80/20 into training/validation, and the other (28 volumes) was used for testing

Information regarding image type, size of the dataset, original and resampled image sizes, proportions used in the data splitting, and image pre-processing steps is described below and summarized in Table 3.

**Nuclei** – The dataset was made available for Kaggle's Data Science Bowl 2018 and consisted of 2D images concerning brightfield and fluorescence microscopy. From the full dataset only 670 images had disclosed annotation ground-truth. Image intensities were rescaled to be in the range of [0-1].

**Colorectal Polyps** – The CVC-ClinicDB dataset was made available as part of the Automatic Polyp Detection in Colonoscopy Videos - Endoscopic Vision Challenge from MICCAI 2015. The dataset, comprised of 612 frames from 29 video sequences, was partitioned at the sequence level to obtain an unbiased estimate of performance. Image intensities were rescaled to be in the range of [0-1] and images were resampled to 192×144.

**Liver** – The dataset was made available for Liver Tumor Segmentation Benchmark (LITS), organized by ISBI 2016 and MICCAI 2017. The public part of the whole dataset with disclosed liver masks, comprised of 131 CT volumes, was used. Although the segmentation is performed at a 2D level, the partitioning was done at the volume/patient level to avoid bias and over-optimistic results. All slices from each 3D CT volume were used. The ground truth labels were comprised of a liver and a tumor segmentation, with only the liver segmentation being used. Before image consumption, images were resampled to 128×128 and intensities were rescaled to the [0-1] range. Intensity clipping was also applied to the interval [-1000; 1000] HU to ensure that the rescaling of intensities was not affected by different bone densities or artifacts from implants.

**Aorta** −The dataset was made available for the Segmentation of THoracic Organs at Risk in CT images (SegTHOR) competition, hosted by ISBI 2019. The public part of the dataset is comprised of 40 CT volumes, with the labelling including heart, aorta, trachea, and esophagus, and only the aorta being considered for our task. Similarly to the liver segmentation task, an intensity clipping was also applied to the interval [-1000; 1000] HU, with the final images being rescaled to the [0-1] range.

## Appendix D. Experimental Configuration

During our experiments, several architectures and sets of hyperparameters were used for both the baselines and the SDN-based models. These are listed in Table 4 and Table 5 respectively. All models were implemented in *Pytorch Lightning* (Falcon, 2019) and trained through the minimization of a combination of dice and cross-entropy as the loss function (Isensee et al., 2018). The Dice score was monitored during training, with the final model being selected through *early-stopping* on the validation set. The optimization of the loss function was stabilized through learning rate annealing, and in order to avoid vanishing and exploding gradients for larger models, gradient clipping and residual connections were used. All other methods followed original descriptions and implementations by the authors of each work, with minor changes being made to Swin-Unet's patch splitting to fit the input size by changing the window size from 7 to 8.

Table 4: Experimental configurations of U-Net and U-Net++.

| **U-Net** | | | | |
|---|---|---|---|---|
| **Segmentation task** | *Nuclei* | *Polyps* | *Liver* | *Aorta* |
| Optimizer | Adam | Adam | Adam | Adam |
| Learning rate | 1.00E-03 | 1.00E-04 | 1.00E-04 | 1.00E-04 |
| Weight decay | 1.00E-05 | 1.00E-05 | 1.00E-05 | 1.00E-05 |
| Convolutional kernel size | 3x3 | 3x3 | 3x3 | 3x3 |
| Activation function | ReLU | ReLU | ReLU | ReLU |
| # layers per scale | 2 | 2 | 2 | 2 |
| Batch normalization | Yes | Yes | Yes | Yes |
| Residual connections | No | Yes | Yes | Yes |
| Batch size per GPU | 20 | 20 | 16 | 16 |
| GPU Model | RTX 2080 Ti | RTX 2080 Ti | RTX 2080 Ti | RTX 2080 Ti |
| Number of GPUs | 4 | 4 | 4 | 4 |
| GPU VRAM | 11 GB | 11 GB | 11 GB | 11 GB |
| Allocated GPU VRAM | 0.7 GB 7 | 2.17 GB | 1.32 GB | 2.56 GB |
| **U-Ne++** | | | | |
| **Segmentation task** | *Nuclei* | *Polyp* | *Liver* | *Aorta* |
| Optimizer | Adam | Adam | Adam | Adam |
| Learning rate | 1.00E-03 | 1.00E-04 | 1.00E-04 | 1.00E-04 |
| Weight decay | 1.00E-05 | 1.00E-05 | 1.00E-05 | 1.00E-05 |
| Convolutional kernel size | 3x3 | 3x3 | 3x3 | 3x3 |
| Activation function | ReLU | ReLU | ReLU | ReLU |
| # layers per scale | 2 | 2 | 2 | 2 |
| Batch normalization | Yes | Yes | Yes | Yes |
| Residual connections | No | Yes | Yes | Yes |
| Batch size per GPU | 20 | 20 | 16 | 16 |
| GPU Model | RTX 2080 Ti | RTX 2080 Ti | RTX 2080 Ti | RTX 2080 Ti |
| Number of GPUs | 4 | 4 | 4 | 1 |
| GPU VRAM | 11 GB | 11 GB | 11 GB | 32 GB |
| Allocated GPU VRAM | 1.92 GB | 6.53 GB | 3.58 GB | 7.08 GB |

Table 5: Experimental configurations of SDU-Net and SDNU-Net.

| | **SDU-Net** | | | |
|---|---|---|---|---|
| **Segmentation task** | *Nuclei* | *Polyps* | *Liver* | *Aorta* |
| Optimizer | Adam | Adam | Adam | Adam |
| Learning rate | 1.00E-03 | 1.00E-04 | 1.00E-04 | 1.00E-04 |
| Weight decay | 1.00E-05 | 1.00E-05 | 1.00E-05 | 1.00E-05 |
| Convolutional kernel size | 3x3 | 3x3 | 3x3 | 3x3 |
| Activation function | ReLU | ReLU | ReLU | ReLU |
| # layers per scale | 2 | 2 | 2 | 2 |
| Batch normalization | Yes | Yes | Yes | Yes |
| Residual connections | No | Yes | Yes | Yes |
| SDN kernel size | $3\times3$ | $3\times3$ | $3\times3$ | $3\times3$ |
| SDN # channels per scale | 100 | 150 | 150 | 150 |
| SDN layers per encoder/decoder | 2 | 1 | 1 | 2 |
| # directions per SDN layer | 2 | 2 | 2 | 2 |
| Directions used | $\downarrow,\rightarrow$ | $\downarrow,\rightarrow$ | $\downarrow,\rightarrow$ | $\uparrow,\downarrow$ |
| Batch size per GPU | 20 | 20 | 16 | 16 |
| GPU Model | RTX 2080 Ti | RTX 2080 Ti | RTX 2080 Ti | V100-SXM2 |
| Number of GPUs | 4 | 4 | 4 | 1 |
| GPU VRAM | 11 GB | 11 GB | 11 GB | 32 GB |
| Allocated GPU VRAM | 1.91 GB | 6.21 GB | 3.30 GB | 6.45 GB |
| | **SDNU-Net** | | | |
| **Segmentation task** | *Nuclei* | *Polyp* | *Liver* | *Aorta* |
| Optimizer | Adam | Adam | Adam | Adam |
| Learning rate | 1.00E-03 | 1.00E-04 | 1.00E-04 | 1.00E-04 |
| Weight decay | 1.00E-05 | 1.00E-05 | 1.00E-05 | 1.00E-05 |
| Convolutional kernel size | 3x3 | 3x3 | 3x3 | 3x3 |
| Activation function | ReLU | ReLU | ReLU | ReLU |
| # layers per scale | 2 | 2 | 2 | 2 |
| Batch normalization | Yes | Yes | Yes | Yes |
| Residual connections | No | Yes | Yes | Yes |
| SDN kernel size | $3\times3$ | $3\times3$ | $3\times3$ | $3\times3$ |
| SDN # channels per scale | 100 | 150 | 150 | 150 |
| SDN layers per encoder/decoder | 2 | 1 | 1 | 2 |
| # directions per SDN layer | 2 | 2 | 2 | 2 |
| Directions used | $\downarrow,\rightarrow$ | $\downarrow,\rightarrow$ | $\downarrow,\rightarrow$ | $\uparrow,\downarrow$ |
| Batch size per GPU | 20 | 20 | 16 | 16 |
| GPU Model | RTX 2080 Ti | RTX 2080 Ti | RTX 2080 Ti | V100-SXM2 |
| Number of GPUs | 4 | 4 | 4 | 1 |
| GPU VRAM | 11 GB | 11 GB | 11 GB | 32 GB |
| Allocated GPU VRAM | 4.63 GB | 10.31 GB | 7.30 GB | 13.21 GB |

## Appendix E. Improvement Summary

In Table 6 we describe the performance gains for SDU-Net and SDNU-Net with respect to each of its baseline models. Positive values in the Dice index indicated a gain in performance, whereas the same is shown through negative values in the Hausdorff distance. All values express a performance gain above the average standard deviation for the considered experiments. The last two columns showcase the average performance gain across both architectures, whereas the last row displays the performance gains averaged across all segmentation tasks. From the summary table we can gather that all the experiments demonstrate an increase in performance when spatial recurrence is included in the segmentation architectures.

The largest increase in performance was observed for the colorectal polyps and liver segmentation tasks, with an average 6.39 and 6.41 Dice increase, and a 1.0 and 0.57 Hausdorff distance decrease, respectively. Arguably this may be due to the variable scales at which polyps and liver cross-sections appear, with large texture changes between both polyps and

Table 6: Performance gain of SDU-Net against U-Net and SDNU-Net against U-Net++, across all segmentation tasks. Highlighted with a underline are values higher than the average standard deviation of the specific group of experiments.

| | SDU-Net | | SDNU-Net | | Dataset Average | |
|---|---|---|---|---|---|---|
| | Dice index ($\uparrow$) | Hausdorff dist. ($\downarrow$) | Dice index ($\uparrow$) | Hausdorff dist. ($\downarrow$) | Dice index ($\uparrow$) | Hausdorff dist. ($\downarrow$) |
| *Nuclei (Caicedo et al., 2019)* | 4.62 | -0.92 | 4.03 | -0.87 | 4.33 | -0.90 |
| *Polyps (Bernal et al., 2017)* | 6.07 | -1.09 | 6.71 | -0.91 | 6.39 | -1.00 |
| *Liver (Bilic et al., 2019)* | 6.87 | -0.49 | 5.94 | -0.35 | 6.41 | -0.42 |
| *Aorta (Lambert et al., 2020)* | 2.37 | -0.37 | 1.17 | -0.38 | 1.77 | -0.38 |
| **Model Average** | 4.98 | -0.72 | 4.46 | -0.63 | 4.72 | -0.68 |

'normal tissue', i.e. non-tumoral, and liver and non-liver tissue. In particular, for the liver segmentation task, it is plausible to claim that spatial positioning plays a larger role, as the organ due to its nature is consistently located adjacent to the same tissues across different patients. Qualitative comparison in Figure 1 also corroborates this assertion. The same argument can be made for the aorta artery segmentation, and improvements were also seen with respect to this task (1.77 Dice increase and 0.61 Hausdorff distance decrease).

## Appendix F. Stability Analysis

The stability of the model's performance was also assessed through averaging results over multiple runs, keeping model selection through early-stopping, and evaluating in a hold-out test set. Each architecture was trained 5 times across different seeds for the batch sampling and weight's initialization, with results summarized in Table 7. The performance was demonstrated to remain consistent across seeds, and only small absolute deviations in the evaluation metrics with respect to the cross-validation study were seen. Overall, the consistent improvement in performance of SDN based architectures can still be observed.

Table 7: Segmentation results averaged across 5 runs with different seeds, for baseline models (U-Net and U-Net++) and SDN models (SDU-Net and SDNU-Net).

| | Dice index ($\uparrow$) | Hausdorff distance ($\downarrow$) | Dice index ($\uparrow$) | Hausdorff distance ($\downarrow$) |
|---|---|---|---|---|
| | *Nuclei (Caicedo et al., 2019)* | | *Liver (Bilic et al., 2019)* | |
| U-Net | 87.14±0.77 | 2.08±0.37 | 85.66±1.31 | 1.82±0.45 |
| U-Net++ | 89.40±0.67 | 1.35±0.33 | 88.09±1.13 | 1.00±0.34 |
| SDU-Net | 91.64±0.55 | 1.01±0.30 | 93.89±1.67 | 0.82±0.41 |
| SDNU-Net | 93.47±0.33 | 0.58±0.22 | 94.78±2.12 | 0.80±0.24 |
| | *Polyps (Bernal et al., 2017)* | | *Aorta (Lambert et al., 2020)* | |
| U-Net | 76.60±1.21 | 3.69±0.64 | 90.68±0.53 | 0.75±0.28 |
| U-Net++ | 78.01±1.33 | 2.57±0.58 | 93.12±0.90 | 0.58±0.40 |
| SDU-Net | 82.76±1.29 | 2.86±0.45 | 93.63±0.66 | 0.50±0.34 |
| SDNU-Net | 86.14±1.48 | 2.33±0.49 | 93.89±0.92 | 0.17±0.13 |

## Appendix G. Model size and computation cost

We also assessed the number of parameters and the computational requirements of our models and their respective baselines, summarized in Table 8. All learnable paramateres included in the computational graph were taken into consideration. The computation of the number of operations in the inference stage was evaluated in multiply-accumulate-operations (MACs) using *ptflops* available at: github.com/sovrasov/flops-counter.pytorch. The scalability of the computational cost with respect to image size was also assessed and is displayed in Table 9. In this setting, both SDU-Net and SDNU-Net architectures were equipped with two SDN layers in both the encoder and decoder, each with 150 channels and sweeps in two directions.

Table 8: Comparison of model size in number of parameters, and inference computation cost in multiply-accumulate-operation (MAC), for SDN-bsed models and respective baselines.

|  | U-Net | U-Net++ | SDU-Net | SDNU-Net |
|---|---|---|---|---|
| **Number of parameters (M)** | | | | |
| *Nuclei* | 1,8 | 2,24 | 5,42 | 6,16 |
| *Polyps* | 7,24 | 9,76 | 10,36 | 11 |
| *Liver* | 7,24 | 9,76 | 10,36 | 11 |
| *Aorta* | 7,24 | 9,76 | 10,36 | 11 |
| **Number of operations (GMACs)** | | | | |
| *Nuclei* | 1,31 | 2,87 | 2,59 | 4,7 |
| *Polyps* | 2,28 | 6,5 | 3,01 | 8,15 |
| *Liver* | 1,71 | 4,87 | 2,36 | 6,09 |
| *Aorta* | 6.08 | 17.33 | 8.03 | 21.67 |

Table 9: Computational cost estimation in multiply-accumulate-operation (MAC) with respect to input size and comparison with baseline methods.

|  | U-Net | U-Net++ | SDU-Net | SDNU-Net |
|---|---|---|---|---|
| **Number of operations (GMACs)** | | | | |
| $3 \times 64 \times 64$ | 0.86 | 2.17 | 1.24 | 2.98 |
| $3 \times 96 \times 96$ | 1.93 | 4.87 | 2.79 | 6.70 |
| $3 \times 128 \times 128$ | 3.44 | 8.86 | 4.96 | 11.92 |
| $3 \times 192 \times 192$ | 7.74 | 19.49 | 11.15 | 26.82 |
| $3 \times 256 \times 256$ | 13.75 | 34.65 | 19.82 | 47.67 |
| $3 \times 384 \times 384$ | 30.95 | 77.96 | 44.60 | 107.27 |

