# OpenReview forum: "Holistic Modeling in Medical Image Segmentation Using Spatial Recurrence"
_MIDL.io/2022/Conference — MIDL 2022_

### Official Review · Reviewer_jkPB · 2022-01-21

**Confidence:** 3
**Preliminary Rating:** 3
**Recommendation:** Poster

**Summary:**

The paper introduces a novel method that incorporates spatial recurrence for medical image segmentation via UNet and U-Net++ networks in order to take into account long-range spatial dependencies in medical data thus improving segmentation performance and enabling holistic modeling. The proposed method is evaluated on four segmentation tasks and demonstrates an increase in segmentation performance relative to U-Net and U-Net++ baselines.

**Strengths:**

The method is well evaluated on 4 segmentation tasks of different modalities and across 3 performance metrics. The authors have done a good job of defining the problems and objectives of the paper.  In addition, the authors have verified the significance and stability of the method through 5-folds cross-validation, multiple random seeding runs, and an ablation study across different model hyper-parameters.

**Weaknesses:**

The main issue with the paper is the unclear presentation and mathematical formulation of the method. The reviewer has found it difficult to understand the methodology, and therefore the true added value of the proposed contribution relative to the already present state-of-the-art methods. Various annotations and variables in Section 2 are not clearly defined especially in eq. 1 and eq. 2 which makes it difficult to valorize the methodological added value of the paper. The reviewers would advise a proper and clearer formulation and demonstration of the method to be done as a condition of acceptance of the paper.

**Deanonymize Review:**

no

**Detailed Comments:**

I am fairly confident that the work is suitable for publication to MIDL and as an extended version for a special issue conditioned on whether the authors address the clarity issues in the methodological formulation and demonstration of the method.

**Final Rating After The Rebuttal:**

4: Weak Accept

**Justification Of The Final Rating:**

The authors have greatly improved their paper based on all the reviewer's questions and feedback. I am overall satisfied with the new paper edition. An open-source code would have been much appreciated.

**Paper Type:**

both

**Questions To Address In The Rebuttal:**

State-of-the-art Comparison:

*The reviewers would appreciate a comparison of the proposed method relative to peer state-of-the-art methods for holistic modeling both at the level of the introduction and the experimentation comparison.

Section 2:
* X \in R^{mnp}: What are M, n , and j , p ?
* Equation 1: Define h, l
* The authors state that "the successive composition of convolutional layers increases the receptive field ". Could the authors provide a thorough explanation on how the sweeps are done? and the intuition behind why they think that successive sweeps can indeed increase the receptive field. A similar question is to be asked regarding: " A further increase is possible by adding the also commonly used pooling layers, leading to an additive increase". What is the intuition behind this ? and why do the authors claim that successive sweeps or pooling layers could increase the receptive fields?
* In the paper, the authors refer to feature matrix and feature maps. If these two are the same, the reviewers would advise a consistent use of terms, else, the reviewers would appreciate that a clear distinction is made between the two.
Section 2.1:
The authors state: "Our proposal is to interleave recurrent sweeps with the convolutional layers. After computing the output O of a convolutional layer, we produce, using recurrent sweeps, another feature map O′ of same dimensions as O, which will be the input for the next layer."
* the reviewers would advise a clarification on how recurrent sweeps are conducted, the concept of recurrent unit g, and the clear definition of i, j, m, and n terms.
* A similar clarity issue persists in the remaining parts of Section 2.





**Special Issue:**

yes

---

### Official Review · Reviewer_uLiQ · 2022-01-24

**Confidence:** 4
**Preliminary Rating:** 4
**Recommendation:** Poster

**Summary:**

This paper introduces a novel spatial dependency layer to introduce spatial recurrence into convolutional neural networks. This is a novel concept which has recently been introduced in an ICLR 2021 paper and used in the domain of image generation (VAEs). In this paper, the layers are used in a segmentation architecture, a U-Net architecture. The paper presents experiments on four different datasets coming from four biomedical image analysis challenges and compares against the U-Net and U-Net++ as baseline methods and presents improvements for all applications.

**Strengths:**

- The paper is in general well written and provides extensive additional information in the supplementary material.
- This paper introduces a novel concept to incorporate global context into convolutional neural networks and show with simple experiments the benefit of the approach. This new concept is interesting to the field.
- Experiments are conducted across four different tasks.

**Weaknesses:**

-  I think the term 'holistic' is not the best; merriam-webster meaning of the word holistic is 'relating to or concerned with complete systems rather than with individual parts'. I would advise the authors to stick with the term spatial recurrence
- I am missing a good introduction and discussion in which the presented approach is put into context with other approaches that try to integrate long-range connections for image segmentation.
- I am concerned about computational cost, both during learning and inference. Appendix D is not linked in the text, but does provide figures on computational cost. The paper would benefit from a short discussion on this.
- This is a 2D approach applied to two 3D tasks, can the authors comment on extension of SDN layers to 3D?


**Deanonymize Review:**

no

**Detailed Comments:**

- No comparison to nnUnet, a pity in my opinion. In addition, the papers reference an older arxiv paper on the nnUnet framework, while a paper has already appeared in Nature Methods: https://www.nature.com/articles/s41592-020-01008-z
- No evaluation on the liver tumor category in the LITS2017 challenge, why not?

**Final Rating After The Rebuttal:**

4: Weak Accept

**Justification Of The Final Rating:**

I want to thank the authors for the extensive rebuttal. I think the paper has improved compared to the original submitted paper. The addition of adding the performance of the nn-Unet framework is appreciated, and it shows that this framework does perform very well on most tasks as well. In my view, the differences are still small, but this approach is interesting to the field. The added paragraph in the introduction with comparisons with other approaches for incorporating global contextual information could be stronger and more extensive in my view. In conclusion, I am sticking with my original rating of this paper.



**Paper Type:**

methodological development

**Questions To Address In The Rebuttal:**

The paper would be strengthened by an improved comparison with other approaches for incorporating global contextual information in the introduction/discussion. Furthermore, a discussion on extensions to 3D and what consequences this has for the computational cost would be of interest for readers. Finally, a baseline performance set by the nnUnet framework would provide another good baseline performance that it well recognized by the field.


**Special Issue:**

no

---

### Official Review · Reviewer_yrL9 · 2022-01-24

**Confidence:** 4
**Preliminary Rating:** 3
**Recommendation:** Poster

**Summary:**

The premise of the work is that current image segmentation architectures lack the ability to learn and use global context. The work therefore integrates prior work on spatial dependency network architecture features into segmentation architectures to enable better inference of global context and evaluates the results on 4 medical imaging datasets. Improved results are seen compared the U-Net and U-Net++ consistently.

**Strengths:**

- Addresses the relevant and important problem of getting more global context into segmentation models.

- Spatial dependency networks have been used for generative models previously, so the application to segmentation is not a large change, but none the less novel.

- The described method shows impressive and consistently good results on multiple datasets in comparison to U-Net and U-Net++.


**Weaknesses:**

- A major weakness of the approach is that it lacks description and discussions of training or inference times or GPU memory usage. Granted some information is given in Appendix C and D, however, C lacks a comparison with the reference U-Net and U-Net++ models and only seems to give the GPU card memory details, not the algorithm usage. Appendix D shows that the proposed models do have larger number of parameters and require more floating point computations, but is still within the same order of magnitude. I would have preferred scalability to be considered in terms of both computational complexity and memory usage.

- No 3D processing is used either in comparison methods or in the proposed method as far as I understand. Considering that this is a medical imaging conference and 3 out of 4 of the datasets considered are in 3 dimensions, this would have been an obvious thing to discuss or try. This relates back to the scalability issue mentioned above. One of the primary challenges in medical imaging is dealing with context across larger and higher dimensional input images. If the proposed method does not scale well to larger images and 3D it is likely not relevant for many problems.

- No mention of newer architectures, such as vision transformers and combinations of vision transformers and CNNs, which attempts to solve similar problems of long range interactions and context.

- I find the SDU-Net explanation unclear. Particular section 2.1 and 2.2. Consider introducing more motivation and reasoning. What is the subscripts i and j indexing over in equation 2?

- Lack of publically available code repository makes reproducing results more difficult.


**Deanonymize Review:**

no

**Detailed Comments:**

- I think the claims that traditional approaches fail at, what the authors call holistic modelling, and in particular the descriptions of why should be expanded on. Why and at what do traditional approaches fail and what is it that the proposed approach is improving on?

- Figure 2 and receptive field points - The receptive field computation illustrated does not seem particularly relevant to me, as most modern neural networks for image processing do not just use convolutions, but also pooling layers, as also mentioned by the authors. The authors say pooling layers leads to an additive increase of (k-1)s in receptive field size, but to my understanding this is wrong. Pooling layers exponentially increase receptive field size.

- "But note that such a receptive may not fully encompass the original image.", I am not sure what the authors mean by this.

- "Also, such composition of layers only weakly models distant pixel relationships across different levels of the architecture. These limitations are unnecessarily restrictive for medical image segmentation.", please be specific here. How is it weak? Why is too restrictive? Moreover, these are unsupported statements on methods that have established themselves through 1000s of publications. Please be specific in your claims, whether done so or not, consider removing them or backing them up with solid references.

- "We found it sufficient to include spatial dependency layers at lower scales of the U-Net++", by what experiments, methods and data?


**Final Rating After The Rebuttal:**

4: Weak Accept

**Justification Of The Final Rating:**

The authors substantially improved the manuscript and added several comparison methods and additional changes. It is a shame that code is not available as is, but it is good that the authors are promising to add a repository.


**Paper Type:**

methodological development

**Questions To Address In The Rebuttal:**

- How does the approach scale with input size in comparison to U-Net and U-Net++? In terms of computation time, training time and memory usage. Empirical and analytical analysis would be relevant.

- How would higher dimensional input affect scalability?

- How does the approach compare to newer architectures based on vision transformers and combinations of vision transformers and CNNs?


**Special Issue:**

no

---

### Meta-Review · Area_Chair_UDvi · 2022-02-14

**Recommendation:** Accept (Poster)
**Confidence:** 5

**Metareview:**

In this paper, a U-Net based segmentation method is proposed, that leverages spatial dependency layer, a new concept which has recently been introduced in an ICLR 2021 paper for image generation. These layers allow to incorporate global context into CNN.
Performance are compared on 4 medical imaging datasets to other state-of-the-art architectures (eg UNet and UNet++ initially).

Strengths:
- This paper introduces a novel concept (proposed elsewhere) into CNN to learn and use global context for image segmentation
- Experimental protocol is sound, presented results are convincing.

Weaknesses:
Initial concern raised by the reviewers focus both on the content and on the form of the papers:
- Comparison to other architectures were lacking (eg Transformers, nnUNet),
- 3D image segmentation was not addressed,
- Computational cost was not properly discussed.
- Some points in the description of the method were not clear

The authors have addressed most of the various questions raised by the reviewers, with additional & results, discussion and re-writing.

Both great, complementary reviews and substantial effort by the authors have led to a significantly improved version of the paper, for which I now recommend acceptance at MIDL.

---

### Decision · Program_Chairs · 2022-02-28

Accept